# AN ANALYSIS OF HUMAN ALIGNMENT OF LATENT DIFFUSION MODELS

**Lorenz Linhardt, Marco Morik, Sidney Bender & Naima Elosegui Borras**
Machine Learning Group, Technische Universität Berlin
Berlin, 10623, Germany
Berlin Institute for the Foundations of Learning and Data – BIFOLD
Berlin, 10586, Germany
{l.linhardt, m.morik, s.bender, n.elosegui.borras}@tu-berlin.de

## ABSTRACT

Diffusion models, trained on large amounts of data, showed remarkable performance for image synthesis. They have high error consistency with humans and low texture bias when used for classification. Furthermore, prior work demonstrated the decomposability of their bottleneck layer representations into semantic directions. In this work, we analyze how well such representations are aligned to human responses on a triplet odd-one-out task. We find that despite the aforementioned observations: **I)** The representational alignment with humans is comparable to that of models trained only on ImageNet-1k. **II)** The most aligned layers of the denoiser U-Net are intermediate layers and not the bottleneck. **III)** Text conditioning greatly improves alignment at high noise levels, hinting at the importance of abstract textual information, especially in the early stage of generation.

## 1 INTRODUCTION

Generative diffusion models have demonstrated remarkable efficacy in image synthesis and editing (e.g. (Dhariwal & Nichol, 2021; Rombach et al., 2022; Ruiz et al., 2023)), image classification (Li et al., 2023a; Clark & Jaini, 2023; Xiang et al., 2023), where they have been shown to make human-like errors and shape bias (Jaini et al., 2024), and in learning object-specific representations (Gal et al., 2023). Finding semantically meaningful internal representations of diffusion models is thus key to better comprehending their aforementioned representations and capabilities. Success in this quest may enable better control over the generation process and yield effective representations in downstream tasks.

Recent findings suggest that the U-Net architectures (Ronneberger et al., 2015), employed as denoisers in most image diffusion models, capture the semantic information in the bottleneck layer ('h-space') (Kwon et al., 2022; Park et al., 2023; Haas et al., 2023). However, the representations generated at medium-depth layers of the up-sampling stage appear to be the most useful for image classification (Xiang et al., 2023) but remain inferior to representations of self-supervised models (Hudson et al., 2023). Despite these insights, the question of where and how diffusion models represent the concepts to be generated remains unsolved.

In this paper, we look at representations of diffusion models from the perspective of human-similarity alignment (Muttenthaler et al., 2023a) (henceforth 'alignment'), as measured on an image-triplet odd-one-out task (Hebart et al., 2020). We hope that this perspective helps us understand generative diffusion models by probing the global structure of representations. As suggested by Sucholutsky et al. (2023), one should measure all components of a model to determine whether it is aligned with a reference system, thus we conduct our evaluation at different layers of the U-Net.

**Contributions**   We contribute to the understanding of diffusion models through an empirical analysis of their representations. For this purpose, we assess their alignment with human similarity judgments and examine the *alignability* of these representations. Our findings reveal that representations from different layers of the U-Net exhibit alignment comparable to classification models trained on much smaller datasets. Notably, the second up-sampling block yields the representa-

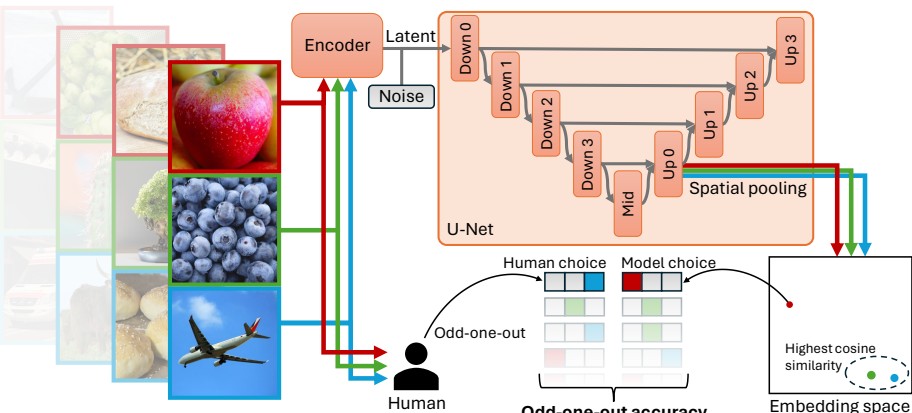

Figure 1: We assess the alignment of image representations obtained from different layers of the U-Net with the human representation space via the triplet odd-one-out task. In this task, three images are presented, and participants identify which image is the least similar to the others. This human judgment is then compared to the model's choice of the odd-one-out based on the cosine similarity of representations.

tions with the highest alignment, from which semantic concepts, except for colors, are also best decodable. We find that alignment decreases with increasing levels of diffusion noise. However, we demonstrate that for high noise levels, text conditioning neutralizes the effect of noise, leading to stable alignment throughout the generative process.

## 2 METHOD

An overview of our workflow for assessing latent diffusion models' alignment with human similarity judgments can be found in Fig. 1. In the following section, we provide details on the individual methodological parts: Sec. 2.1 describes how representations are extracted, Sec. 2.2 contains details on how their alignment is measured, and in Sec. 2.3 additional information on the improvement of alignment is provided. In contrast to other works on semantic spaces in diffusion models (e.g. Kwon et al. (2022); Park et al. (2023); Haas et al. (2023)), our focus is on Stable Diffusion (SD) models (Rombach et al., 2022) due to their training on large and diverse datasets, presumably leading to rich representations.

### 2.1 REPRESENTATION EXTRACTION

To extract the representations from diffusion models, we follow the approach of Xiang et al. (2023). Given an image $x$ and noise level $t$, we feed the denoising network $f_\theta(z_t, t, c)$ a noisy latent $z_t$, generated using the latent diffusion encoder, and optionally some text embedding $c$. We denote the noise level as the percentage of total noising steps $T$ taken, where the exact amount of noise is determined by the scheduler [1] (see Appx. B for a visualization). We then record the internal representation of the U-Net after each of its constituent blocks separately. We apply average pooling to the spatial dimensions to obtain our final (zero-shot) representations per layer $r_t^l$ (see Appx. E for a comparison to alternatives).

### 2.2 REPRESENTATIONAL ALIGNMENT WITH HUMANS

To quantify the extent of representational alignment between humans and diffusion models, we follow Muttenthaler et al. (2023a) and use the THINGS dataset, which consists of neuroimaging and behavioral data of 4.70 million unique triplet responses, crowdsourced from 12,340 human participants for $m = 1854$ natural object images (Hebart et al., 2020) and builds on the THINGS

---

[1]We use the default scheduler for each model from the diffusers library `https://github.com/huggingface/diffusers`.

database (Hebart et al., 2019). To create the THINGS dataset, humans were given a triplet odd-one-out task, consisting of discerning the most different element in a set of three images belonging to distinct object types. There is no correct choice and for any given triplet the answer may vary across participants. The odd-one-out accuracy (OOOA) is a metric used to quantify model and human alignment by assessing what fraction of the odd-one-out determined via the network's representations corresponds to the image selected by humans. The similarity matrix $\boldsymbol{S} \in \mathbb{R}^{m \times m}$ of the model's representations is computed by $\boldsymbol{S}_{a,b} := \boldsymbol{r}_a^T \boldsymbol{r}_b / (\|\boldsymbol{r}_a\|_2 \|\boldsymbol{r}_b\|_2)$, i.e. the cosine similarity between the representations extracted from the model $f_\theta$. For a triplet $\{i, j, k\} \in \mathcal{T}$, where $\mathcal{T}$ is the set of all triplets and w.l.o.g. $\{i, j\}$ are the indices of the most similar pair of the triplet, according to the human choice:

$$\text{OOOA}(\boldsymbol{S}, \mathcal{T}) = \frac{1}{|\mathcal{T}|} \sum_{\{i,j,k\} \in \mathcal{T}} \mathbb{1}[(\boldsymbol{S}_{i,j} > \boldsymbol{S}_{i,k}) \wedge (\boldsymbol{S}_{i,j} > \boldsymbol{S}_{j,k})] \tag{1}$$

### 2.3 Alignment by Affine Probing

Poor alignment does not mean that the relevant concepts are not contained in the representations. It has been shown that a linear transformation can drastically improve the OOOA (Muttenthaler et al., 2023a). Thus, in addition to measuring the *zero-shot* alignment of representations extracted from diffusion models (i.e. without modifying the representations), we measure their affine *alignabilty*, i.e. how much their OOOA can be increased using an affine transformation. For this step, we follow Muttenthaler et al. (2023a;b) and learn a *naive transform*, i.e. a square weight matrix $\boldsymbol{W}$ and bias $\boldsymbol{b}$ for each set of representations:

$$\underset{\boldsymbol{W}, \boldsymbol{b}}{\arg \min} \; -\frac{1}{|\mathcal{T}|} \sum_{\{i,j,k\} \in \mathcal{T}} \log \left( \frac{\exp(\hat{\boldsymbol{S}}_{i,j})}{\exp(\hat{\boldsymbol{S}}_{i,j}) + \exp(\hat{\boldsymbol{S}}_{i,k}) + \exp(\hat{\boldsymbol{S}}_{j,k})} \right) + \lambda \|\boldsymbol{W}\|_F^2. \tag{2}$$

Here, $\hat{\boldsymbol{S}}$ is the cosine similarity matrix of the transformed representations $\tilde{\boldsymbol{r}} = \boldsymbol{W} \boldsymbol{r} + \boldsymbol{b}$. Intuitively, the goal of the optimization is to maximize the relative similarity $\hat{\boldsymbol{S}}_{i,k}$ of the images not chosen as the odd-one-out by the human participants. The magnitude of the transformation is kept small by the regularization term, in order not to distort the original representations too much. We use 3-fold cross-validation (CV) on the THINGS dataset and pick the best $\lambda \in \{10^i\}_{i=-4}^1$. The resulting 'probed' representations can then be evaluated in the same way as the original ones.

## 3 Experiments

We evaluate three latent diffusion models (Rombach et al., 2022) trained on the LAION-5B dataset (Schuhmann et al., 2022): Stable Diffusion 1.5[2] (SD1.5), Stable Diffusion 2.1[3] (SD2.1), and Stable Diffusion Turbo[4] (SDT), the latter being an adversarial distilled version of SD2, enabling generation with fewer steps (Sauer et al., 2023). The main body of the paper focuses on SD2.1, and we refer to the appendix for results obtained from the other models. First, we analyze how well the representations of the diffusion models are aligned with human similarity judgments. Then we show how the alignment of diffusion model representations varies over noise levels and the different layers. Lastly, we show the influence of text-conditioning on the alignment.

### 3.1 How Well Aligned are the Representations of Diffusion Models?

We first analyze the representations generated from $\boldsymbol{x}$ without further text conditioning. This is the most naive and perhaps faithful implementation of the image triplet tasks, as only image information is used. In Fig. 2, it can be seen that the highest OOOA across layers is 45.31% for SD1.5, 45.47% SDT, and 43.29% for SD2.1. These values are below the average of the models evaluated by Muttenthaler et al. (2023a) and roughly comparable to self-supervised models trained on ImageNet-1k. Note that due to choice disagreement between humans, the maximum achievable accuracy is only

---

[2] https://huggingface.co/runwayml/stable-diffusion-v1-5
[3] https://huggingface.co/stabilityai/stable-diffusion-2-1
[4] https://huggingface.co/stabilityai/sd-turbo

| Model | Zero-Shot | Probing |
|---|---|---|
| ViT-B-32[†] | 42.52% | 49.69% |
| SimCLR[†] | 47.28% | 56.37% |
| CLIP$_{Image}^{†}$ | 47.64% | 61.07% |
| CLIP$_{Text}$ | 48.47% | 57.38% |
| ResNet50[†] | 49.44% | 53.72% |
| AlexNet[†] | 50.47% | 53.48% |
| VGG-16[†] | 52.09% | 55.86% |
| SD2.1 | 43.29% | 54.48% |
| SD2.1$_{Cond}$ | 44.02% | 57.24% |
| SD1.5 | 45.31% | 56.29% |
| SDT | 45.47% | 55.60% |

Figure 2: **Left**: Comparison of the OOOA from the best layer of the diffusion model to models analysed by Muttenthaler et al. (2023a) (†). **Middle/Right**: OOOA per layer and noise level for SD2.1 **without** or **with** text conditioning, respectively. The alignment of SD2.1 is highest at the second up-sampling block (i.e. 'Up 1'). It is within the lower range of OOOAs observed for models trained on ImageNet-1k. After probing, SD2.1 is more aligned than unimodal self-supervised models or classifiers. Also, label-conditioning (Cond) improves alignment, especially at high noise levels.

67.22% $\pm$ 1.04% Hebart et al. (2020), whereas the accuracy of random guessing is around $33.\dot{3}\%$. We conclude that the capabilities of SD models are not reflected in the human alignment of their intermediate representations.

### 3.1.1 Can the Representation be Aligned Easily?

In this section, we briefly present the OOOA results obtained after applying an affine transformation, learned for each block individually, as outlined in Sec. 2.3. It can be seen in Fig. 6 that the overall pattern across layers and noise levels does not change, but alignment increases generally. While this improvement is substantial, the alignment of the transformed representations is only slightly better than that of models trained on much less data (Muttenthaler et al., 2023a), after a similar transformation. This may indicate either that the dimensions relevant for human similarity judgments are not much better represented in SD models, or that more flexible transformations are needed to extract them.

### 3.2 How Does Alignment Vary Across Layers?

In unconditional diffusion models, the bottleneck layer of the U-Net appears to carry the most semantic information (Kwon et al., 2022) and to encode concepts as directions. This idea is further supported by recent works (Park et al., 2023; Haas et al., 2023). We find that this does not hold for SD models.

The OOOA obtained from the representations extracted at different layers and for different levels of noise are displayed in Fig. 2. The most aligned layers are the intermediate up-sampling layers, which corresponds to the layers found to be most useful for linear classification (Xiang et al., 2023), albeit we find little to no degradation until noise levels of at least 30%. Furthermore, one might assume that for small $t$, the model would not need to involve the deeper layers to remove the little noise that is left and thus the representations at the deeper layers degrade. This does not appear to be the case.

We speculate that the reason for the discrepancy with the results previously reported on unconditional diffusion models lies in the complexity of the SD models, which were trained on a diverse dataset with various modes. Here, the learned representation might not admit simple linear extraction of concepts.

### 3.2.1 Do Layers Encode Different Concepts?

A natural question to ask is whether different human concepts are represented at different levels of depth in SD models, for example, more abstract concepts being more salient in deeper layers. To investigate this question, we make use of the VICE dimensions (Muttenthaler et al., 2022), which

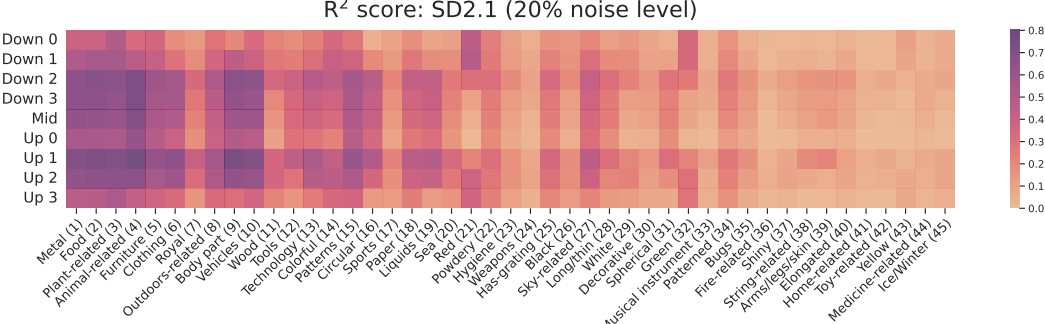

Figure 3: Per-concept $R^2$-scores for the regression of VICE dimensions from SD2.1 representations, measured at different U-Net blocks for a noise level of 20%. Colors tend to be decodable at shallower layers, whereas most other concepts peak at the second up-sampling block.

model the human similarity space using a human-interpretable positive orthogonal basis. Using VICE, each image of the THINGS dataset can be decomposed into 45 dimensions. We use the labeling of the dimensions from (Muttenthaler et al., 2023a), noting that it is only a post-hoc interpretation of their semantics.

We follow the experimental protocol of Muttenthaler et al. (2023a) and train a multinomial ridge regression to predict the VICE dimensions from the extracted representations. The results were obtained using 5-fold cross-validation, where, within each fold, the regularization parameter was chosen from $\{10^i\}_{i=-2}^5$ using leave-one-out CV.

In Fig. 3 the regression metric, measured by $R^2$ is computed for distinct concepts at varying layer depths. Qualitatively, it can be observed that except for the colors red, green, and yellow, which follow the same pattern of correlation, there is little differentiation of concepts across layers. Most concepts are best decodable from the second up-sampling block. See Appendix C.2 for additional concept-wise results across noise levels. Most concepts remain stable up to about 40% noise and degrade beyond that.

### 3.3    What is the Impact of Text-Conditioning on Alignment?

Diffusion models are often trained and used with textual prompts to guide generation. In this section, we investigate the effect of textual conditioning of SD models on their alignment. In particular, we condition the reconstruction of $x$ from $z_t$ on '*a photo of a* $\langle \text{OBJ} \rangle$', where $\langle \text{OBJ} \rangle$ is replaced by the name of the object depicted in the image, as per the image file name.

We observe that textual conditioning stabilizes alignment across noise levels, keeping the variability across layers intact but reducing the variability across noise levels to a low level. At very high levels of noise, where the denoiser has to rely almost exclusively on the text conditioning, there may even be improvements to the OOOA, stemming from the relatively higher text-embedding OOOA (see Appx. D). For SD2.1, especially the bottleneck and adjacent blocks benefit from text conditioning beyond their unconditional maximum values, although only at higher noise levels. Improvements are less localized in SD1.5. We refer to Appx. D for the full set of results as well as a comparison with conditioning on the output of a text captioning model.

## 4    Conclusion

Despite previous work uncovering semantic directions in smaller diffusion models and the outstanding capabilities of stable diffusion models, we show that internal representations of the latter are not exceedingly aligned with the similarity space extracted from human behavioral experiments. While an affine transformation improves alignment significantly, the gap to contrastive image-text models trained on large amounts of data remains unclosed. This suggests that diffusion models trained on large multi-modal datasets do not have a linearly decodable representation space. Of the various

blocks of the denoising network, we find the intermediate up-sampling blocks yield the most aligned representations. Furthermore, we observe that conditioning the denoising on textual object labels improves alignment at high levels of noise.

The presented results open several lines of future investigations. Does the residual structure of the U-Net architecture itself affect the alignment of its individual components? Is the visual reconstruction objective of generative models orthogonal to human alignment of representations? Perhaps the way the representations are structured even requires a different measure of alignment (e.g. evaluating the triplet task with a similarity measure other than cosine similarity). As the representation space might be highly non-linear, alignment-increasing transformations may need to allow for non-linearity.

### ACKNOWLEDGMENTS

LL, MM, and NEB gratefully acknowledge funding from the German Federal Ministry of Education and Research under the grant BIFOLD24B, SB from BASLEARN—TU Berlin/BASF Joint Laboratory, co-financed by TU Berlin and BASF SE.

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

## A    RELATED WORK

Denoising diffusion models have emerged as effective generative models for a variety of tasks, including unconditional image generation (Sohl-Dickstein et al., 2015; Ho et al., 2020; Dhariwal & Nichol, 2021), text-to-image synthesis (Ho & Salimans, 2021; Saharia et al., 2022; Rombach et al., 2022), and inverse problems (Song et al., 2021; Chung et al., 2022). As these models gain widespread adoption, understanding their internal representations becomes crucial. Their text-to-image synthesis capabilities suggest semantic knowledge, which has proven useful for classification (Li et al., 2023a; Jaini et al., 2024) and learning representations for downstream tasks (Mittal et al., 2023). Analyzing the representation space facilitates the identification of failure modes (Liu et al., 2024) and semantic directions (Haas et al., 2023; Park et al., 2023). Such analysis, akin to work on GANs (Härkönen et al., 2020), also allows for the manipulation at the bottleneck layer of U-Net (Kwon et al., 2022). A parallel line of inquiry attempts to train diffusion models specifically for representation learning (Hudson et al., 2023; Mittal et al., 2023) or to infuse their representations with concepts (Ismail et al., 2023).

The comparison of behavior between neural networks and humans has been approached from different angles: the majority consider error consistency in image classification (Geirhos et al., 2020; 2021; Rajalingham et al., 2018), others focus on semantic similarity judgments (Jozwik et al., 2023; Peterson et al., 2018; Aminoff et al., 2022; Marjieh et al., 2022), or analyse perceptual similarity (Zhang et al., 2018; Jagadeesh & Gardner, 2022). We build upon an analysis of human and neural network similarity judgments Muttenthaler et al. (2023a) to assess the alignment of representations extracted from pretrained diffusion models.

## B    VISUALIZATION OF NOISE LEVELS

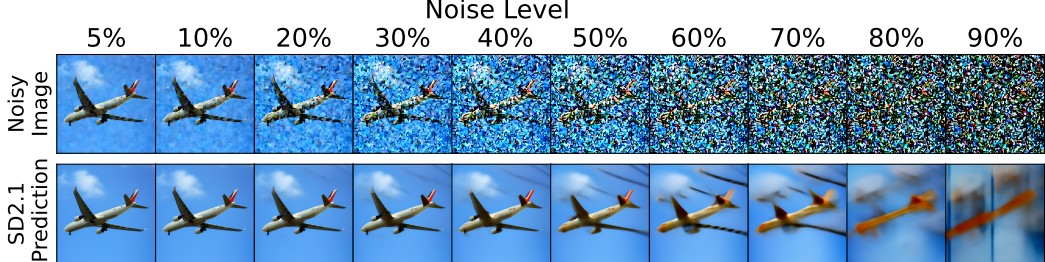

Figure 4: **Top**: The decoded latents for different noise levels. **Bottom**: The images $x$ reconstructed from the noisy latents via a single forward step by SD2.1.

Fig. 4 shows both the noisy latent and its $x$ reconstruction for Stable Diffusion 2.1. The reconstruction quality remains good up to 60% noise, while from 80% noise on, the image is barely identifiable. This matches the decrease in alignment observed in representation space.

## C    ADDITIONAL RESULTS FOR UNCONDITIONAL IMAGE REPRESENTATIONS

In this section we report the OOOA results for unconditioned representations, using all evaluated SD models. The patterns discernable in Fig. 5 follow a similar pattern as described in Sec. 3.1, but in SD1.5 OOOA is almost as high at the middle layer as it is at the second up-sampling block.

### C.1    ADDITIONAL PROBING RESULTS

The complete OOOA results for affine transformed representations, using all models, are reported in Fig. 6. The general pattern is consistent across models and similar to the one observed for the original representations, albeit at a generally higher level of alignment. Specifically, we see that the Up 1 block yields the most aligned representations, with slightly lower values at its symmetric counterpart, Down 2. For SD1.5, the layers between those two layers are more aligned than in SD2.1 and SDT.

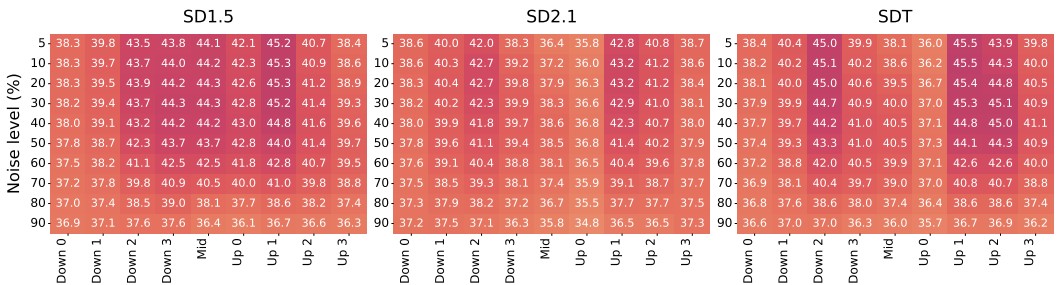

Figure 5: Odd-one-out accuracy for **zero-shot** representations **without** text conditioning. Intermediate up-sampling layers are most aligned with human similarity judgments.

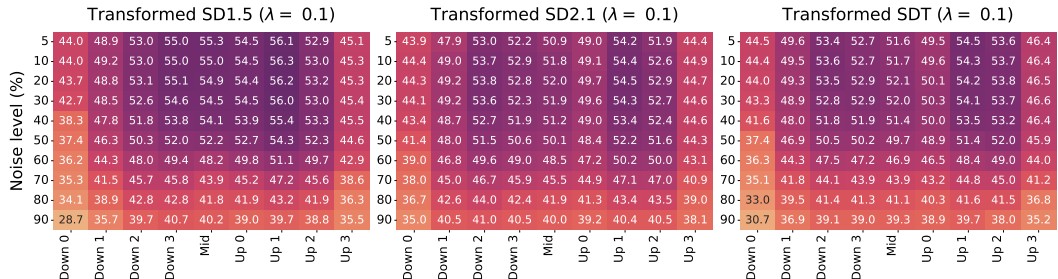

Figure 6: Odd-one-out accuracy for **transformed** representations **without** text conditioning. The observed alignment is greatly improved over zero-shot representations (Fig. 5).

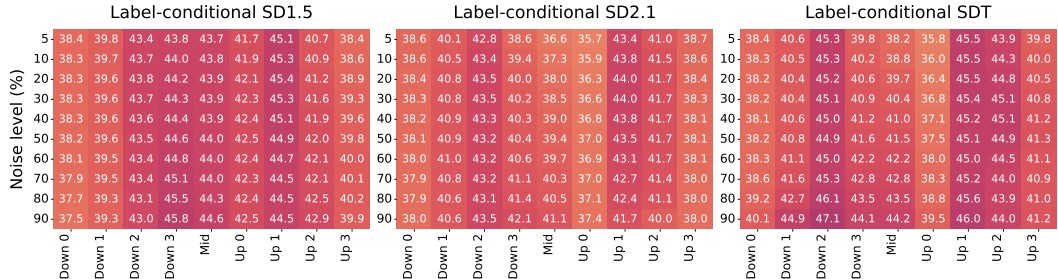

Figure 7: Odd-one-out accuracy for **zero-shot** representations **with** text conditioning on the label ('*a photo of a* ⟨OBJ⟩'). The observed alignment is increased at higher noise levels.

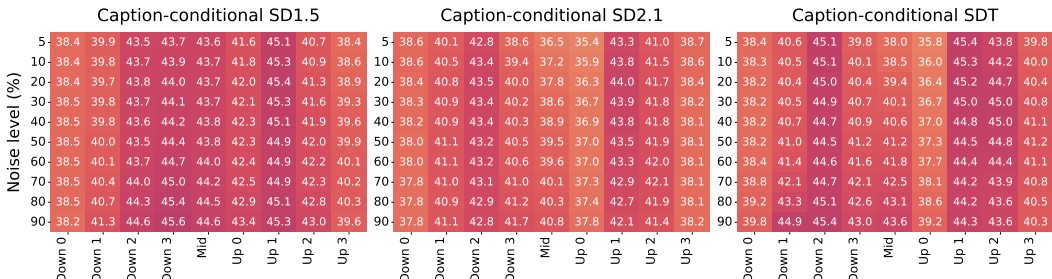

Figure 8: Odd-one-out accuracy for **zero-shot** representations **with** text conditioning on an image caption generated by a captioning model. The observed alignment is comparable with conditioning on the label (Fig. 7).

At Down 0 with high noise, OOOA values can get below the random-guessing level of $\frac{1}{3}$. This is due to counting a triplet-task solution as wrong if more than one pair shares the highest similarity value and thus the representations do not unambiguously yield an odd-one-out.

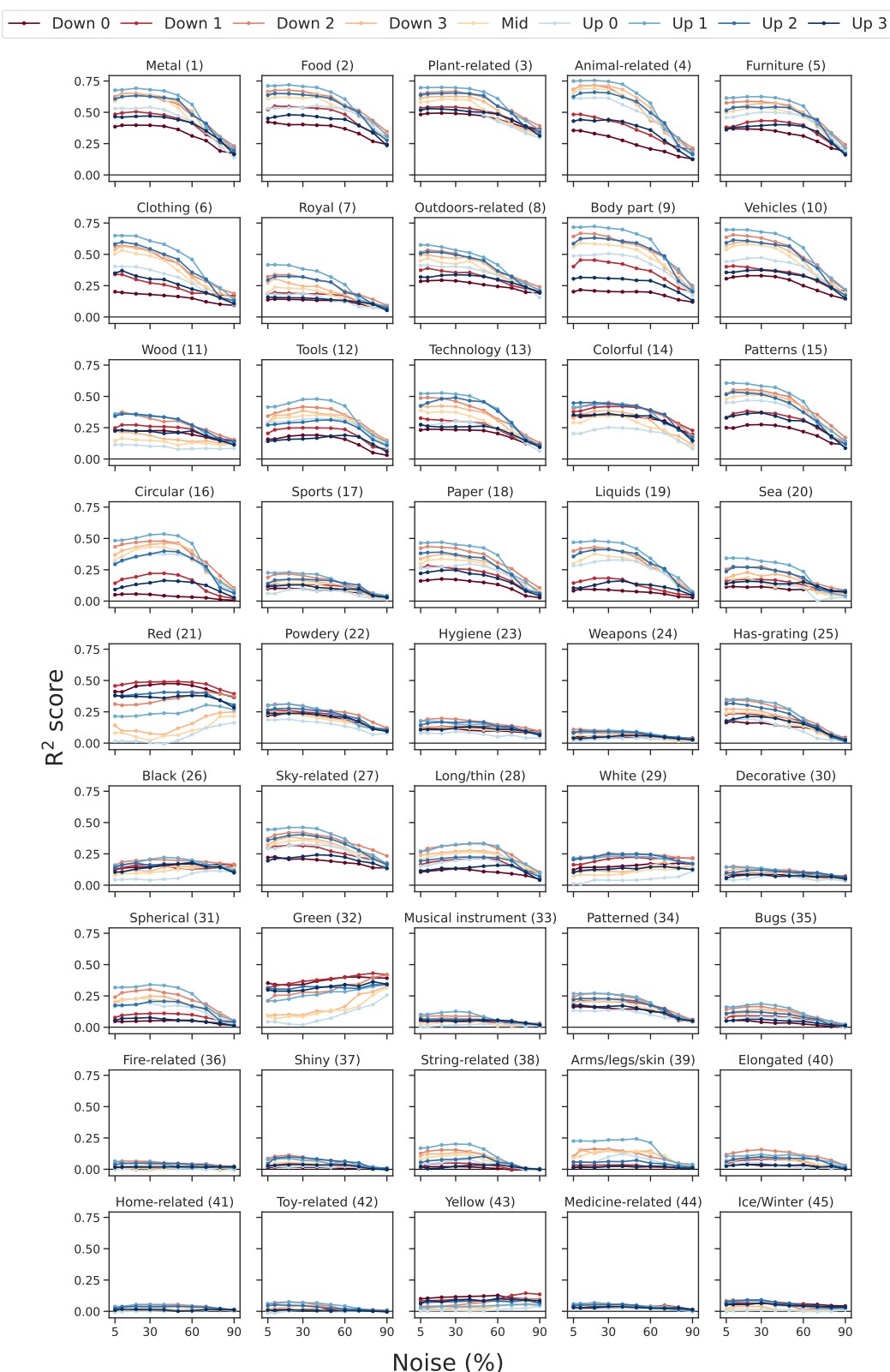

Figure 9: Regression $R^2$ scores for SD2.1 for all blocks and various noise levels.

## C.2 Per-Concept Analysis

In Fig. 9, we present the concept-wise regression scores for representations obtained from unconditional denoising, over different layers and levels of noise. Generally, higher noise levels degrade the decodability of concepts, although small improvements can be seen up to about 30% noise for some concepts. Exceptions are the 'circular' and 'string-related' dimensions, which improve up to 40% noise, and the 'green' and 'yellow' dimensions, which see small improvements up to 80% noise. Interestingly, the inner representations (Down 3, Mid, Up 0) increasingly represent color dimensions (like 'green', 'red') for noise levels higher than 50%. This indicates that color information is only relevant for these layers in the early steps of the diffusion process.

## D   Additional Results for Text-Conditional Image Representations

In this section we report the OOOA results for text-conditional representations, using all evaluated SD models. Fig. 7 contains the results for object-label-conditioned denoising, and Fig. 8 for caption-conditioned denoising. For the latter, we used a BLIP (Li et al., 2022) image captioning model from the LAVIS library (Li et al., 2023b). Exact label information does not seem to be necessary, as the results obtained from the caption-conditioned model are very similar. Furthermore, our observations indicate that the text embedding has a stronger impact on the distilled model Stable Diffusion Turbo (SDT), particularly when the noise level is high. This aligns with expectations, considering that this model is specifically optimized for single-step inference from complete noise.

As a reference, we report the OOOA of the text embeddings of the object labels: 44.30% for SD1.5, and 48.47% for SD2.1 and SDT. Here, we make use of the text encoders used to train the SD models and only take the last non-padding token of the embedded text, which has been found to contain most information (Ding et al., 2023).

## E   Dimensionality Reduction

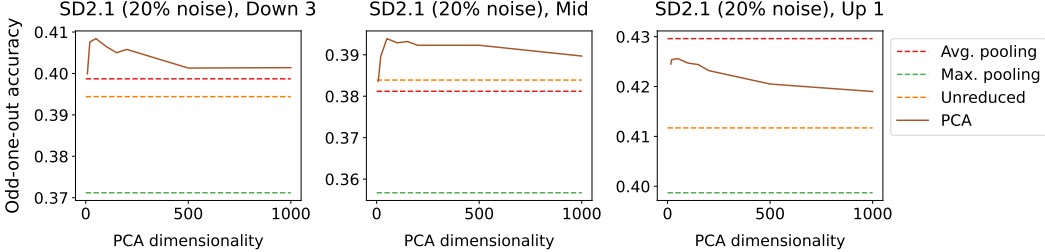

Figure 10: Comparison of different strategies for reducing representation dimensionality for SD2.1.

While pooling is necessary to achieve reasonably sized representations, it may discard relevant information. Here, we briefly evaluate alternatives to average pooling the spatial dimensions of the extracted representations. Specifically, for selected layers, we compare the OOOA of unpooled, max-pooled, average-pooled, and PCA-reduced representations. For efficiency reasons, we evaluate OOOA on a subset of 1,000,000 triplets. Fig. 10 shows that indeed, average pooling, as also employed by previous work (e.g. (Xiang et al., 2023)) is more favorable than max pooling and better than or on par with unpooled representations- There is no dominating dimensionality reduction strategy when comparing to PCA. While PCA-based dimensionality reduction generally leads to small improvements in alignment over unpooled evaluation, we observe that these come almost exclusively from centering the data.

