# OpenReview forum: "An Analysis of Human Alignment of Latent Diffusion Models"
_ICLR.cc/2024/Workshop/Re-Align — ICLR 2024 Workshop Re-Align Poster_

### Official Review · Reviewer_veLK · 2024-02-16
**Adding human-alignment angle to diffusion models**

**Rating:** 2
**Fit:** 3
**Confidence:** 2

**Workshop Review:**

# Summary
The paper analyzes the alignment of latent diffusion models with human responses on a triplet odd-one-out task. It identifies that the most aligned layers of the denoiser U-Net are intermediate upsampling layers, not the bottleneck, as suggested by prior works. They show that text-conditioning improves alignment at higher noise-levels.
The paper highlights that despite the outstanding capabilities of diffusion models, their internal representations are not highly aligned with human similarity space. This suggests that diffusion models may not have a linearly decodable representation space.
# Strong points
-	Well written and easy to follow, motivation and line of experiments is very clear
-	Interesting and novel results
# Weak points
-	Some of the results are hard to understand (e.g. in Figure 2, zero-shot and probing are nor clearly defined before)
-  The link between plot labels and caption is missing (intermediate up-sampling block = up1?)
-  Context if achieved accuracies are high or low is missing (what is baseline, max accuracy?)
# Clarity
Clear structure and writing, easy to follow. Results are well structured and follow three main experiments, with clear goals.
# Correctness
Methods and experimental design seem well-grounded in previous works, while still adding a novel research angle. Everything is well described.
# Novelty
No novel methods, but novel research question adding a human-alignment angle to the semantic latent space discovery/analysis in diffusion models. Results challenge previous works and suggest not only looking at the bottleneck layer but also upsampling layers for semantic spaces.
# Interest to community
Potential interest for machine learning researchers interested in diffusion models or interpretability of models. Limited interest for neuroscientists and cognitive scientists.
# Recommendation
I recommend to accept this paper, as it is fitting to the workshop and of potential interest for the community. The findings are novel and expand on previous works by adding an human-alignment angle.
# Recommendations for improvement
-	Main results and conclusion can be formulated clearer and provide some context and potential future implications.
-	Clearer figure captions (e.g. intermediate layer = up1,2), it takes some time to know where to look
-	For the future, it might be interesting to look at different human-alignment tasks

**Reason For Not Giving Higher Score:**

While the work is interesting and provides a new angle, I doubt it is of greater interest to all to justify a talk.

**Reason For Not Giving Lower Score:**

Work is well-suited for the workshop

**Reviewer Domain:**

machine learning

---

### Official Review · Reviewer_vf3j · 2024-02-19
**Interesting paper with many surprising findings**

**Rating:** 3
**Fit:** 3
**Confidence:** 2

**Workshop Review:**

The authors investigate the representation alignment between humans and diffusion models. I found this paper particularly interesting because it has several surprising, unexpected observations: (1) stable diffusion is not as aligned as other less well performing generative models. This seems to imply that stable diffusion develops a strategy to represent images that is orthogonal to humans. (2) The bottleneck is not the informative/aligned layer, but rather some up-sampling layers are more aligned with human representations. (3) representations are not linearly decodable for complex datasets; (4) text conditions can improve representation alignment even in the high noise regime. This may imply that human's visual representations are also closely related to language?

This paper is clearly written and sound. My only nitpick is that the study is not very mechanistic, i.e., making a bunch of observations without making very mechanistic explanations.

**Reason For Not Giving Higher Score:**

There's no higher score.

**Reason For Not Giving Lower Score:**

I actually ride on fence whether to give a 2 or a 3. My reluctance was coming from the fact that this paper is basically an experimental paper without mechanistic explanations (i.e., does not provide constructive suggestions on e.g., how to construct better generative models, or more human-aligned models). Said that, I ended up giving a 3 since I was probably asking too much for a short workshop paper.

**Reviewer Domain:**

machine learning

---

### Official Review · Reviewer_tzfZ · 2024-02-23
**Good paper but limited novelty**

**Rating:** 2
**Fit:** 3
**Confidence:** 2

**Workshop Review:**

This work investigates the alignment between internal representations in Stable Diffusion models and the similarity space derived from human behavioral experiments. Despite the effectiveness of stable diffusion models, the authors found that their internal representations did not closely match the human behavioral similarity space.

**Clarity** and **Correctness**: The paper is well-written, and the pipeline illustration is straightforward and intuitive. However, Section 2.4, *Alignment by Affine Probing* could be improved. It is unclear what the optimization's objective is, requiring reading the referenced papers for a better understanding.

**Novelty**: The novelty of the paper is somewhat limited. It concentrates on analyzing Stable SD models within the same setting used by other papers studying different models (e.g., CLIP-based and ResNet). Despite the distinct nature of these models—given that existing works rely on classifiers rather than diffusion models—the method for extracting representations from the diffusion models is not novel but rather follows an existing approach.

**The paper is well-suited for the workshop.**

**Reason For Not Giving Higher Score:**

The novelty of the paper is somewhat limited.

**Reason For Not Giving Lower Score:**

The paper is well-written and easy to follow. The authors proposed several future investigations that could be interesting to explore.

**Reviewer Domain:**

machine learning

---

### Author Response · Authors · 2024-03-13
**Reply to reviewers**

We thank the reviewers for the positive feedback as well as for the suggested improvements.

Following reviewer $\textit{tzfZ}$, we added clarifications to the probing optimization's objective to convey the underlying intuition.
Furthermore, we tried to address the ambiguities pointed out by reviewer $\textit{veLK}$, in particular unclear elements in the figure captions, and emphasized the min/max achievable accuracy.
We also thank reviewer $\textit{vf3j}$ for their constructive review; we hope to soon be able to extend our findings with explanations of the mechanisms underlying our observations.

---

### Decision · Program_Chairs · 2024-03-02

Accept (Poster)